# Sex–Gender Differences in Adult Coeliac Disease at Diagnosis and Gluten-Free-Diet Follow-Up

**DOI:** 10.3390/nu14153192

**Published:** 2022-08-04

**Authors:** Gloria Galli, Giulia Amici, Laura Conti, Edith Lahner, Bruno Annibale, Marilia Carabotti

**Affiliations:** Medical-Surgical Department of Clinical Sciences and Translational Medicine, Sant’Andrea University Hospital, Sapienza University of Rome, 00185 Rome, Italy

**Keywords:** coeliac disease, gender, small intestine, gastrointestinal symptoms, females, males

## Abstract

Coeliac disease (CD) is an immune-mediated enteropathy triggered by gluten ingestion. At CD diagnosis, gender differences have been previously reported, but data regarding follow-up are scant. We investigated gender differences in CD adult patients both at the time of diagnosis and at follow-up after the start of the gluten-free diet (GFD). This is a longitudinal cohort study on adult CD patients diagnosed between 2008 and 2019. Clinical, biochemical, and histological data were assessed and compared between males and females. At diagnosis, female gender was significantly associated with signs of malabsorption (OR 3.39; 95% CI: 1.4–7.9), longer duration of symptoms and/or signs before the diagnosis (OR 3.39; 95% CI: 1.5–7.5), heartburn (OR 2.99; 95% CI: 1.1–8.0), dyspepsia (OR 2.70; 95% CI: 1.1–6.5), nausea/vomit (OR 3.53; 95% CI: 1.1–10.9), and constipation (OR 4.84; 95% CI: 1.2–19.6) and less frequently associated to higher body mass index (OR 0.88; 95% CI: 0.8–0.9) and osteopenia/osteoporosis (OR 0.30; 95% CI: 0.1–0.7) compared to male patients. After 12–30 months, females presented lower median BMI, performed less frequently histological control, and had more frequently anaemia and hypoferritinaemia compared to males. No significant differences concerning the presence of gastrointestinal symptoms, adherence to GFD, and Marsh score were found. Gender differences found at CD diagnosis mostly disappear at the follow-up, showing that these differences can be solved over time.

## 1. Introduction

In the last years, the debate on gender- and sex-related differences has been gaining importance in several diseases, above all autoimmune ones. Both gender and sex could influence the prevalence, severity and expression of autoimmune diseases. The main reasons for sex differences are genetic and hormonal, but gut microbiota also plays an important role in this topic. In addition, the gender-specific perception and manifestation of signs/symptoms also related to men/women′s diverse lifestyles contribute to these differences [1,2,3]. 

Coeliac disease (CD) is an immune-mediated enteropathy triggered by gluten ingestion in genetically predisposed subjects [4]. The start of the gluten-free diet (GFD) is the only known treatment able to ameliorate gastrointestinal symptoms and/or malabsorptive signs and restore duodenal mucosal atrophy in a heterogeneous range of CD patients [4,5]. Factors influencing adherence to GFD are widely studied [6]. One of the most important factors is the adherence to the scheduled follow-up visits, which, if inadequate, could be responsible for scarce GFD compliance and, consequently, symptoms and/or signs persistence [6,7]. 

CD is more frequent in the female gender, with a female–male ratio of about 2–2.5:1 [8]. Previous studies aimed to understand gender differences at the time of CD diagnosis in both adults and children showed conflicting results. Some authors reported a more frequent “atypical” CD presentation in male patients [9,10], while others did not find differences in symptoms presentation between genders [11,12]. Contradictory results have also been found concerning the delay in CD diagnosis, with few studies showing a more advanced age at the time of CD diagnosis in males with respect to females [13,14,15] and others showing no significant differences between genders [9]. Concerning biochemical alterations, anaemia has been more frequently reported in newly diagnosed CD females [12], while other few studies reported only minor disparities between female and male gender in CD [11]. 

However, to our knowledge, no studies reported how those differences modify after the start of the GFD in the adult CD patients’ follow-up. 

This study aims to evaluate sex and gender differences in adult CD patients both at the time of diagnosis and at the GFD follow-up in order to understand if those disparities persist after the start of the GFD or could influence its adherence.

## 2. Materials and Methods

### 2.1. Study Design 

This is a longitudinal cohort study focusing on adult CD patients diagnosed between 2008 and 2019 at our academic tertiary referral centre (Sant’Andrea University Hospital of Rome). 

We considered the following inclusion criteria: (1) adult patients (≥18 years old); (2) CD diagnosis based on positive CD-specific serology (anti-transglutaminase IgA and/or anti-endomysium IgA autoantibodies, normal total IgA) and concomitant duodenal villous atrophy classified as Marsh 3A–C damage; (3) complete patient’s information with a structured questionnaire including personal, clinical data and concomitant diseases reported at CD diagnosis and GFD follow-up visits. 

Patients performed a baseline evaluation at the CD diagnosis and follow-up outpatient visits at least 12 months after CD diagnosis. Demographic, clinical, biochemical, and histological data were collected at the time of CD diagnosis and follow-up. The time span between the onset of GI symptoms and/or malabsorption signs leading to the diagnosis and the CD diagnosis was assessed at the baseline visit. After at least 12 months from the start of the GFD, we suggested an endoscopic/histologic re-evaluation of the duodenal mucosa. 

Biological sex is genetically determined, while gender regards the social context [1,2]. In the current study, biological sex and gender overlapped in all patients. The term “gender” was used for sociodemographic and lifestyle features, such as age, symptoms, comorbidities, body mass index, and family history, while “sex” was used for immunologic and genetic features.

Patients’ data were anonymised to guarantee the secure processing of sensitive data and collected into a predefined spreadsheet. The study was conducted according to the Sapienza Sant’Andrea Hospital protocol, and written informed consent was obtained from all included patients at the time of CD diagnosis. The study protocol conforms to the ethical guidelines of the 1975 Declaration of Helsinki, as reflected in a priori approval by the institution’s human research committee. 

#### Endoscopic Procedures and Histological Classification

Esophagogastroduodenoscopy (EGDs) with at least four biopsies obtained from the second part of the duodenum was performed in all patients using a flexible video-gastro scope (Olympus GIF-Q165, GIFQ185). A subgroup of CD patients underwent endoscopic histologic re-evaluation in a period ranging from 12 to 30 months after beginning the GFD.

The same pathologist expert in the field of CD, who was blinded to clinical data, examined the intestinal biopsies for each patient both at diagnosis and follow-up. Biopsies were analysed after haematoxylin and eosin and immunohistochemical staining for CD3 counts and were assessed using the Marsh classification system modified by Oberhuber [16,17]. The histological persistence of the above-listed alterations was described and classified as Marsh 1, Marsh 2, or Marsh 3 (A, B, or C), as previously defined. 

### 2.2. Serology, CD-Specific Antibodies, and Nutritional Evaluations 

Anti-transglutaminase (tTG) and anti-endomysium (EMA) IgA antibodies were assessed in all patients both at diagnosis and at the time of the follow-up outpatient visit. 

IgA tTG antibodies were assayed using an enzyme-linked immunosorbent assay kit commercially available from Eurospital (Trieste, Italy). An indirect immunofluorescence assay was used to detect IgA EMA in monkey oesophageal sections. Other blood assays, such as complete blood cell count, ferritin, total cholesterol, triglycerides, and total protein count, were performed using standard laboratory techniques and analysed according to the normal ranges expected from the male and female gender. In particular, according to the World Health Organization, anaemia was diagnosed when haemoglobin level was <13 g/dL in males and <12 g/dL in females; hypoferritinaemia when ferritin level was <24 ng/mL in males and <11 ng/mL in females; hypocholesterolaemia when cholesterol level was <110 mg/dL in both males and females; hypotriglyceridaemia when triglycerides level was <50 mg/dL in both males and females and hypoproteinaemia when total protein level was <6.4 g/dL in both males and females [18].

Bone mineral density (BMD) of the lumbar spine and left femoral neck was assessed in the included patients by dual-energy X-ray absorptiometry (DEXA) within 3 months of CD diagnosis. The BMD values were expressed as standard deviation scores that compared the individual BMD measurements to those of young adults (T-score). Based on the World Health Organization criteria, T-scores between −1.0 and−2.4 indicated osteopenia, and scores of −2.5 or less indicated osteoporosis in both men and women [19].

Weight and height were also obtained to determine the body mass index (BMI). According to the World Health Organization, normal weight was defined as BMI ≥ 18.5 Kg/m^2^, while underweight was defined as a BMI below this limit in both genders. 

### 2.3. GI Symptoms and GFD Adherence Assessment

The presence of GI symptoms was assessed both at diagnosis and at the follow-up, through a standardised questionnaire currently used in our department, including the Bristol stool chart [20,21,22].

Upper GI symptoms, such as vomiting/nausea, heartburn, regurgitation, dysphagia, and troublesome dyspepsia, were considered if they were present at least once a week for at least the last 3 months [23,24]. Lower GI symptoms, such as abdominal pain and troublesome bloating, were considered if they were present at least once a week; constipation was defined as fewer than 3 spontaneous bowel movements per week or straining, with lumpy, hard stools (Bristol scale 1–2); and diarrhoea was defined as increased frequency of bowel movements (>3 stools/day) or decreased consistency of stool (loose or liquid stools, Bristol scale 6–7) for at least 3 months before the CD diagnosis [25]. Adherence to GFD was assessed by the five-point validated Biagi score [26], consisting of four questions about how patients managed their GFD (0–2 = voluntary gluten ingestion, not adequate GFD; 3–4 = adequate GFD); this score was administered by two dedicated physicians during follow-up visits. Both at the diagnosis and at the follow-up visits, before the GFD adherence assessment, patients were also instructed and specifically interviewed by the two dedicated physicians to rule out gluten occult contaminations.

### 2.4. Statistical Analyses

Descriptive statistics are expressed as numbers, percentages (%) of totals, and medians (ranges). Univariate analyses were performed by *t*-test, Fisher’s exact test for continuous or categorical variables to identify differences between male and female CD patients both at the diagnosis and at the time of clinical re-evaluation. Odds ratios (ORs) and 95% confidence intervals (CIs) were used to identify variables related to the dependent variable of interest (female gender) and were obtained by multivariate logistic regression analysis. Age at CD diagnosis (as a continuous variable), BMI (as a continuous variable), duration of symptoms and/or signs before CD diagnosis (≥3 years), associated non-autoimmune comorbidities, presence of GI symptoms (and also specifically heartburn, nausea/vomit, dyspepsia, abdominal pain, and constipation), signs of malabsorption, and presence of osteopenia/osteoporosis at the time of the CD diagnosis were included in the logistic model. 

A subgroup analysis was also performed by Fisher’s exact test to compare CD pre-menopausal and post-menopausal female patients.

Two-tailed *p* values < 0.05 were considered statistically significant. Statistical analyses were performed by MedCalc© Statistical (MedCalc Software, Ostend, Belgium, version 12.7.8).

## 3. Results

### 3.1. Celiac Disease Patients at Diagnosis

In total, 76 (19.3%) out of 393 patients with CD diagnosis between 2008 and 2019 were excluded since they did not meet the inclusion criteria (Figure 1).

Of the 317 included patients, 87 (27.5%) were males and 230 (72.5%) females. Differences between male and female patients at the CD diagnosis are reported in Table 1.

At the univariate analysis, we found that several features distinguished female from male patients: female patients were significantly younger and had a lower median BMI value, with otherwise no significant differences regarding the proportion of underweight patients. Females reported more frequently both GI symptoms and signs of malabsorption (in particular anaemia and hypoferritinaemia) with a more frequent longer duration (>3 years) of symptoms/signs before the CD diagnosis compared to male patients. Interestingly, the presence of osteopenia/osteoporosis was more present in male patients.

Some of these sex and gender differences were confirmed in the multivariate logistic regression analysis (Table 2). In detail, the female gender had a 3.39 times higher probability with respect to the male gender to exhibit a longer duration of symptoms/signs before the CD diagnosis and to present signs of malabsorption. Regarding GI symptoms, the multivariate analysis showed a higher probability for females to complain of upper GI symptoms such as nausea/vomit (OR 3.53; 95% CI 1.14–10.93), heartburn (OR 2.99; 95% CI 1.11–8.05), dyspepsia (OR 2.70; 95% CI 1.12–6.46), and constipation (OR 4.84; 95% CI 1.19–19.63) compared to male patients. Conversely, osteopenia/osteoporosis and lower BMI were found to be significantly more represented in male respect female patients.

To assess the possible impact of menopause on CD presentation, we compare pre-menopausal (*n* = 198) and post-menopausal (*n* = 32) female patients (Appendix A). Interestingly, post-menopausal patients report greater latency between symptoms/signs onset and CD diagnosis (*p* = 0.02). As expected, post-menopausal female patients were older (*p* < 0.0001) and presented more frequently with osteopenia/osteoporosis compared to pre-menopausal female patients (*p* = 0.01). No differences were found regarding signs of malabsorption and histological findings.

### 3.2. Celiac Disease Patients at the GFD Follow-Up

Two hundred thirty-six patients (74.4% of the total; males 27.1% and females 72.9%) performed the follow-up visit at least after 1 year of GFD, while 81 patients (25.5%) were lost at the follow-up (Figure 1).

Compared to the included CD patients, those lost at the follow-up were not significantly different with respect to the male-to-female ratios (28.4% males and 71.6% females; *p* = 0.77) and median age (males: 45 years (range 18–72 years), females: 35 years (range 18–65 years) *p* = 0.61).

Table 3 reports personal, clinical, and biochemical features of CD male and female patients at the GFD follow-up. In univariate analysis, female patients presented a lower median BMI (*p* = 0.01) and performed less frequently endoscopic/histological control after the start of the GFD (*p* = 0.02) than with respect to male patients. No differences were found concerning median months of GFD, compliance with the GFD, GI symptoms, antibodies positivity, and Marsh score at the histological control. Concerning signs of malabsorption, female patients presented more frequently anaemia and hypoferritinaemia, with the latter persisting in one-third of female CD patients after about 14 months of GFD (*p* < 0.01). However, no differences were found concerning Marsh 3 persistence at the histological control between females with and without hypoferritinaemia (*p* = 0.165) (data not shown).

## 4. Discussion

The purpose of this study was to assess the presence of sex and gender differences in CD patients both at the time of diagnosis and at the GFD follow-up. Interestingly, we found that male and female patients presented several differences at the time of the CD diagnosis, which disappeared almost completely after the start of the GFD.

At the time of CD diagnosis, we found that female patients presented more frequently with GI symptoms respect to male patients. In particular, upper GI symptoms (dyspepsia, nausea/vomit, and heartburn) and constipation were strictly associated with the female gender in our cohort. Previous studies reported the presence of more ‘classical’ symptoms in CD women differently from ‘non-classical’ or more silent symptoms shown in CD men [11,12,14]. Some studies reported a higher rate of dyspepsia and anaemia in CD females [11,14], while others described a higher prevalence of malabsorptive signs and hypertransaminasaemia in male patients [9,10].

Differences between females′ and males′ GI symptoms are also described in the general population. For example, it is well known that the prevalence of functional GI disorders such as irritable bowel syndrome (IBS) with constipation and functional dyspepsia is higher in women [27,28]. On the contrary, diarrhoea-predominant IBS is slightly prevalent in men [27,29]. Concerning other upper GI symptoms, differences between genders are less pronounced, with a rate of gastroesophageal reflux similar in both females and males [29,30,31]. It is important to underline that gender-related different perceptions of symptoms cannot be excluded.

In addition, female patients presented a longer duration of symptoms-signs (≥3 years) before the CD diagnosis, highlighting a risk of diagnostic delay compared to male patients. The higher frequency of “atypical” CD symptoms such as upper GI symptoms and constipation in our cohort of CD female patients could have contributed to the diagnostic delay. We can also hypothesise that women get used to living with these symptoms, perceived as chronic and not serious; similarly, iron deficiency with or without anaemia could not be appropriately taken into account, by both the patients and the physicians, due to the common presence in the general female population and attributed to the physiological blood loss with the menstrual cycle in pre-menopausal women [32]. Regarding the diagnostic delay, discordant results have been previously reported. On the one hand, some authors reported a diagnostic delay in the male gender due to the non-classical and more subtle symptoms complained before the CD diagnosis [7,13,15]. On the other hand, some studies showed a shorter duration of the presenting symptoms in CD male patients respect females, leading them to early undergo diagnostic tests [9,33,34].

Interestingly, male patients presented a significantly higher rate of osteopenia/osteoporosis at the time of CD diagnosis, detected by the DEXA examination. This result, apparently in contrast with the well-known risk factor for altered BMD associated with aged females in the general population, has been previously reported in a study carried out by our teamwork [35] and confirmed by a more recent study [36]. Even if the mechanism is poorly known, it can be likely due to a selective malabsorptive phenotype or to an altered hormonal profile in men CD patients.

Regarding the concomitant presence of comorbidities, we found that male CD patients presented more frequently non-autoimmune comorbidities (i.e., cardiovascular, metabolic, endocrinological) compared to female patients (*p* = 0,03) even if not confirmed by the logistic regression analysis. On the other hand, we found a similar prevalence of autoimmune comorbidities between male and female CD patients; this trend has been similarly reported by other authors [9,11] and might be likely ascribed to the genetic autoimmune predisposition of CD patients acting as a background for the development of other autoimmune diseases [37].

The subgroup analysis conducted on pre-menopausal and post-menopausal CD females has not shown unexpected results confirming the broad similarities between these two populations (Appendix A). In fact, both the higher age and the more frequent rate of osteopenia/osteoporosis observed are expected in post-menopausal patients. Concerning the longer duration of symptoms/signs (≥3 years) before the CD diagnosis reported in post-menopausal females, it is possible to hypothesise that, also according to the higher age presented, these patients underestimated their symptoms/signs for years making possible the CD diagnosis only after a long time.

Concerning the presence of sex and gender differences in CD patients at the GFD follow-up, we found that the majority of the differences described at CD diagnosis disappeared after a median of 14–15 months of GFD.

Regarding the presence of GI symptoms after GFD, we reported a significant improvement of symptoms during the follow-up, with no significant differences regarding symptoms prevalence between males and females. This is in line with the positive effect of GFD [5] that, in our cohort, seems to occur after a few months regardless of the prevalence of upper and lower GI symptoms and the gender of the patients. We can speculate that females, having more symptoms at diagnosis compared to males, showed a greater clinical improvement.

Interestingly, we found that adherence to the GFD and to the follow-up outpatient visits was not influenced by gender being similar between males and females. Few data on gender and adherence to the outpatient visits follow-up are available with heterogeneous follow-up times, reporting no significant differences between CD males and females [7,38].

Available data concerning GFD adherence found conflicting results [39]: some studies have not shown significant differences between the two genders [40,41]; in two studies, CD male patients seem to exhibit slightly poorer adherence to the GFD than women [42,43], while another study demonstrated a better GFD adherence in males respect females [44]. Due to the well-known limits of traditional questionnaires for GFD assessment, new methods to evaluate GFD adherence, such as faecal/urinary gluten peptides measurement, have recently been developed [45]. In our study, we tried to improve the reliability of the GFD adherence assessment by a careful interview performed by dedicated physicians combined with the use of the validated Biagi score for each patient [26]. As a future perspective, additional long-term data will be needed in order to confirm these results after a longer follow-up and by adding new methods to measure GFD adherence.

Although there are no gender differences, in our cohort of adult coeliac patients, the adherence to the GFD and follow-up outpatient visits are slightly higher with respect to those reported in Literature. In fact, we found a mean percentage of GFD adherence in both males and females of 92.8% respect to 42–91% reported in other studies, while compliance to the outpatient follow-up visits was 74.4% in both genders with respect to 62–73% of previous studies [7,46]. We can theorise that being followed in a tertiary CD referral centre could have encouraged adherence to the follow-up visits and to the GFD.

Regarding the histological control after GFD, no differences were found between female and male patients’ sex in the rate of both histological healing and atrophy persistence. Previous studies aimed to understand the rate of histological healing in CD patients after different timing of GFD adherence have not found differences between sex and genders in the histological recovery [47,48,49,50]. One study conducted in a large population of both coeliac adults and children with a heterogeneous range of follow-up periods (2–5 years from the CD diagnosis) documented a mildly increased risk of persistent villous atrophy in male patients (OR 1.43; 95% CI 1.07–1.90) [51]. On the contrary, we found that male patients adhered significantly more often to histological control compared to females. Similarly to our study, Haere P. et al. demonstrated a male prevalence in the endoscopic/histologic follow-up [52]. Once again, we can speculate that female CD patients tend to minimise symptoms and give little importance to their state of health.

Two other variables distinguished CD males and females during the GFD follow-up. First, similarly to the time of CD diagnosis, female patients have a lower median BMI value compared to male patients (22.2 vs. 23.9 Kg/m^2^, respectively). Some authors reported a significantly lower BMI in female patients with respect to males at the CD diagnosis [9,10,11], but no data on the follow-up are available. In our cohort, female CD patients showed a lower BMI compared to male patients, likely due to the short-term follow-up. Further long-term follow-up studies are needed to evaluate these differences in the long term. It is important to underline that no differences were found with respect to the proportions of underweight patients between males and females, meaning that the differences found with respect to the mean value of BMI are not clinically significant.

Second, concerning signs of malabsorption, hypoferritinaemia was the only biochemical alteration that distinguished males from females at the GFD follow-up. This result seems likely ascribable to two factors: first, ferritin represents the storage of iron and takes a long time to restore; second, pre-menopausal women have a monthly menstrual cycle that, periodically, leads to iron loss, especially in some individuals with concurrent polymenorrhea or menometrorrhagia [53].

We are aware of some limitations of this study, first of all, the short duration of the GFD follow-up. However, we aimed to focalise on a short-term follow-up in order to better evaluate GI symptoms and malabsorptive signs modification. The specific evaluation of sex and gender differences in the long-term follow-up could be interesting but fall outside of this study. Secondly, since the inclusion of patients covers a long-time period, the lack of a statistical analysis evaluating the possible time effect on the studied population could be a limit. Though, since 2008, the features analysed during the outpatient visits have been almost the same, including the histologic control after GFD.

To the best of our knowledge, this is the first study aimed to investigate sex and gender differences both at CD diagnosis and at GFD follow-up in a prospective cohort of adult patients.

## 5. Conclusions

In conclusion, this study demonstrated that sex and gender differences found at CD diagnosis mostly disappear at the GFD follow-up, showing that these differences can be solved over time. Being followed in a referral centre could have encouraged adherence to the follow-up visits and to the GFD, representing a management model able to adequately treat this life-long condition.

## Figures and Tables

**Figure 1 nutrients-14-03192-f001:**
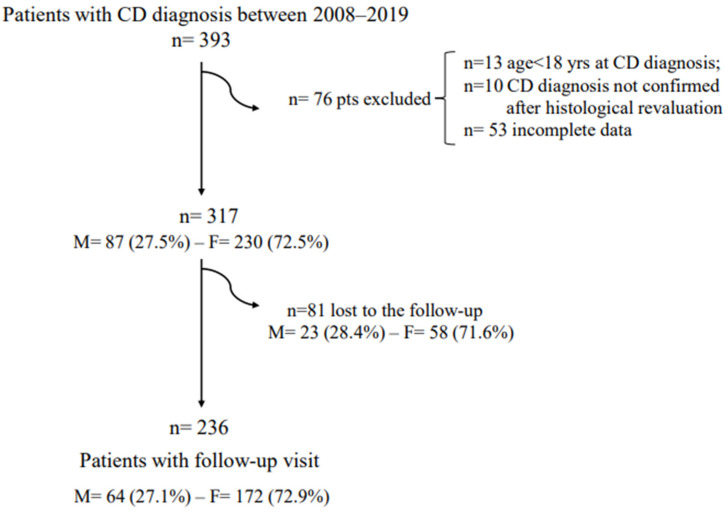
Flowchart of the included patients.

**Table 1 nutrients-14-03192-t001:** Features and comparison between male and female patients at coeliac disease diagnosis time.

	Male Patients	Female Patients	*p*
	*n* = 87	*n* = 230	
**Median age, years (range)**	43 (18–72)	36 (18–76)	** *0.001* **
**Median BMI * kg/m^2^ (range)**	23.5 (15.9–35.7)	21.6 (16.1–38.2)	** *0.001* **
**Underweight (<18 kg/m^2^)**	9.5%	15.0%	0.26
**Comorbidities**			
Autoimmune	31.0%	30.4%	1
Other #	25.3%	14.8%	** *0.03* **
**1st degree family history of CD §**	21.8%	17.4%	0.41
**≥3 years duration of symptoms/signs before the CD diagnosis**	21.1%	45.4%	** *<0.0001* **
**tTg IgA Ab** ≥ **10 ULN**	40.5%	51.7%	0.09
**Clinical presentation** Classical symptoms	30.2%	22.2%	0.14
**GI ° symptoms**			
Total of pts with GI symptoms	62.1%	85.0%	** *<0.001* **
Nausea/vomit	10.3%	25.6%	** *0.003* **
Heartburn	19.5%	31.7%	** *0.03* **
Regurgitation	12.6%	22.0%	0.07
Dysphagia	6.9%	10.1%	0.51
Postprandial fullness/early satiety	24.1%	47.6%	** *<0.001* **
Abdominal pain	34.5%	52.4%	** *0.005* **
Abdominal bloating	39.1%	61.7%	** *<0.001* **
Constipation	3.4%	20.7%	** *<0.0001* **
Diarrhoea	27.6%	24.7%	0.66
**Signs of malabsorption**	50.0%	72.5%	** *0.02* **
Anaemia	16.0%	48.4%	** *<0.0001* **
Hypoferritinaemia	25.0%	63.9%	** *<0.0001* **
Hypocholesterolaemia	15.0%	8.0%	** *0.01* **
Hypotriglyceridaemia	1.0%	10.3%	** *0.01* **
Hypoproteinaemia	9.0%	6.0%	0.12
**Osteopenia/osteoporosis**	66.2%	49.2%	** *<0.01* **
**Marsh 3C at diagnosis time**	51.7%	56.9%	0.44

*** BMI**—body mass index; **# Other**—comorbidities more common—metabolic, cardiovascular; **§ CD**—coeliac disease; **° GI**—gastrointestinal.

**Table 2 nutrients-14-03192-t002:** Multivariate logistic regression. Variables associated with the female gender (dependent variable) at the time of CD diagnosis.

	Odds Ratio	95% CI	*p*
**Age #**	0.80	0.37–1.68	0.55
**Body mass index #**	0.88	0.79–0.98	** *0.02* **
**Non-autoimmune comorbidities §**	0.42	0.16–1.09	0.07
**>3 years duration of symptoms/signs before CD diagnosis**	3.39	1.52–7.56	** *0.002* **
**Gastrointestinal symptoms ***	0.70	0.20–2.34	0.56
**Nausea/vomit**	3.53	1.14–10.93	** *0.02* **
**Heartburn**	2.99	1.11–8.05	** *0.02* **
**Dyspepsia**	2.70	1.12–6.46	** *0.02* **
**Abdominal pain**	0.98	0.39–2.44	0.98
**Abdominal bloating**	2.22	0.92–5.32	0.07
**Constipation**	4.84	1.19–19.63	** *0.02* **
**Signs of malabsorption ***	3.39	1.44–7.97	** *0.005* **
**Osteopenia/osteoporosis**	0.30	0.13–0.72	** *0.007* **

# Continuous variable; § comorbidities more common = Metabolic, Cardiovascular; * at least one symptoms/sign.

**Table 3 nutrients-14-03192-t003:** Features and comparison between male and female patients at the time of the GFD follow-up.

	Male Patients	Female Patients	*p*-Value
	*n* = 64	*n* = 172	
**Median months of GFD * (range)**	15 (12–30)	14 (12–30)	0.32
**Adequate GFD**	92.2%	93.4%	0.77
**Median BMI § kg/m^2^ (range)**	23.9 (16–32.8)	22.2 (16.6–36.6)	** *0.01* **
**Underweight (BMI < 18 kg/m^2^)**	3.3%	4.5%	1
**GI ° symptoms**			
Total of pts with GI symptoms	31.2%	32.1%	1
-Nausea/vomit	0	3.0%	
-Heartburn	10.9%	5.9%	0.25
-Regurgitation	6.2%	3.6%	0.46
-Dysphagia	1.5%	0.6%	0.47
-Postprandial fullness/early satiety	3.2%	6.5%	0.52
-Abdominal pain	9.4%	15.5%	0.28
-Abdominal bloating	14.1%	16.7%	0.84
-Constipation	9.4%	9.0%	1
-Diarrhoea	12.5%	5.4%	0.08
**Antibodies positivity**	18.0%	24.8%	0.37
**Signs of malabsorption**	23.4%	34.5%	0.115
Anaemia	6.8%	16.3%	0.055
Hypoferritinaemia	14.7%	33.3%	** *<0.01* **
Hypocholesterolaemia	5.0%	1.8%	0.346
Hypotriglyceridaemia	5.0%	11.6%	0.22
Hypoproteinaemia	1.8%	2.7%	1
**Histological control**	93.7%	81.4%	** *0.02* **
**Marsh 3 (atrophic disease)**	30.0%	28.6%	0.86
**Marsh score**			
Marsh 0	63.3%	55.0%	0.39
Marsh 1	8.3%	15.0%	0.25
Marsh 2	0	2.9%	
Marsh 3A	21.6%	18.6%	0.69
Marsh 3B	5.0%	7.1%	0.75
Marsh 3C	1.6%	1.4%	1

* GFD—gluten-free diet; § BMI—body mass index; ° GI—gastrointestinal.

## Data Availability

The data presented in this study are available on request from the corresponding author. The data are not publicly available due to ethical restrictions.

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
