# Peer review of "Sex–Gender Differences in Adult Coeliac Disease at Diagnosis and Gluten-Free-Diet Follow-Up"

_nutrients, 2022, doi:10.3390/nu14153192_

Round 1

Reviewer 1 Report

This paper is an exhaustive evaluate of sex and gender differences in adult CD patients both at the time of diagnosis and at the GFD follow-up. Though, in this work have been used the methods for evaluating the adherence to a GFD are comprised of a dietary questionnaire, serological test, or clinical symptoms; however, none of these methods generate either a direct or an accurate measure of dietary adherence. A small-bowel biopsy is the “gold-standard” method for CD diagnosis. However, according to most clinical guidelines, its role in the follow-up of patients is limited to cases involving a lack of clinical response or the recurrence of symptoms. There is currently a tool to measure gluten immunogenic peptides (GIP) and monitor the gluten-free diet in the follow-up of celiac patients. Therefore, this point should be included in the discussion of the paper as a future perspective in new clinical trials (Wieser, H.; Ruiz-Carnicer, Á.; Segura, V.; Comino, I.; Sousa, C. Challenges of Monitoring the Gluten-Free Diet Adherence in the Management and Follow-Up of Patients with Celiac Disease. Nutrients 2021, 13, 2274. https://doi.org/10.3390/nu13072274).

I would like to raise some comments from the manuscript:

·        The abstract should be a total of about 200 words maximum.

·        Spaces are missing in the abstract.

·        Line 18 uses an abbreviation not previously described in the abstract.

·        Also, I think the authors mean 'adherence to GFD´ rather than to 'GFD compliance' 'adherence to GFD´.

·        Line 217: The paragraph should go below table 3. Check this part.

·        English should be unified. For example, coeliac disease and celiac disease.

·        References should be described as follows in Journal Articles. For example, in bibliography 2, the month of publication must be removed. Also, it would be convenient to add all the doi.

1. Author 1, A.B.; Author 2, C.D. Title of the article. Abbreviated Journal Name Year, Volume, page range.

Author Response

Open Review – REVIEWER 1

(x) I would not like to sign my review report

( ) I would like to sign my review report

English language and style

( ) Extensive editing of English language and style required

( ) Moderate English changes required

( ) English language and style are fine/minor spell check required

(x) I don't feel qualified to judge about the English language and style

Yes         Can be improved             Must be improved          Not applicable

Does the introduction provide sufficient background and include all relevant references?

(x)          ( )           ( )            ( )

Are all the cited references relevant to the research?

( )           (x)          ( )            ( )

Is the research design appropriate?

( )           (x)          ( )            ( )

Are the methods adequately described?

(x)          ( )           ( )            ( )

Are the results clearly presented?

(x)          ( )           ( )            ( )

Are the conclusions supported by the results?

( )           (x)          ( )            ( )

Comments and Suggestions for Authors

This paper is an exhaustive evaluate of sex and gender differences in adult CD patients both at the time of diagnosis and at the GFD follow-up. Though, in this work have been used the methods for evaluating the adherence to a GFD are comprised of a dietary questionnaire, serological test, or clinical symptoms; however, none of these methods generate either a direct or an accurate measure of dietary adherence. A small-bowel biopsy is the “gold-standard” method for CD diagnosis. However, according to most clinical guidelines, its role in the follow-up of patients is limited to cases involving a lack of clinical response or the recurrence of symptoms. There is currently a tool to measure gluten immunogenic peptides (GIP) and monitor the gluten-free diet in the follow-up of celiac patients. Therefore, this point should be included in the discussion of the paper as a future perspective in new clinical trials (Wieser, H.; Ruiz-Carnicer, Á.; Segura, V.; Comino, I.; Sousa, C. Challenges of Monitoring the Gluten-Free Diet Adherence in the Management and Follow-Up of Patients with Celiac Disease. Nutrients 2021, 13, 2274. https://doi.org/10.3390/nu13072274).

We thank the Reviewer for the time spent and the attention paid to analyze our work.

We agree with the Reviewer regarding the importance given in the last years to the GIPs in monitoring the GFD. Unfortunately, in our center we started to use it only in patients with unclear results at the time of follow-up (for example: discordance between histology and GFD adherence evaluation) from 2020 making unsuitable to insert this method in this study. However, it is important to underline that we currently use the Biagi score which is a validated questionnaire to evaluate adherence to  GFD assessment.

As suggested, to discuss this point  we added a new paragraph in the discussion (Discussion: lines 306-312)

I would like to raise some comments from the manuscript:

  • The abstract should be a total of about 200 words maximum.

As suggested, we have modified the abstract in order to include a maximum of 200 words 

  • Spaces are missing in the abstract.

As suggested, we have modified the abstract in order to add the missing spaces.

  • Line 18 uses an abbreviation not previously described in the abstract.

As suggested, we have modified the abbreviation (BMI) using the full name.

  • Also, I think the authors mean 'adherence to GFD´ rather than to 'GFD compliance' 'adherence to GFD´.

We thank the Reviewer for this suggestion. We replaced “GFD compliance” with “Adherence to GFD”  in the whole text (changes highlighted).

  • Line 217: The paragraph should go below table 3. Check this part.

We modified the paragraph as suggested. 

  • English should be unified. For example, coeliac disease and celiac disease.

Thanks for the comment. We opportunely unified English language.

  • References should be described as follows in Journal Articles. For example, in bibliography 2, the month of publication must be removed. Also, it would be convenient to add all the doi.
  1. Author 1, A.B.; Author 2, C.D. Title of the article. Abbreviated Journal Name Year, Volume, page range.

We thank the Reviewer for his attention. We fixed the bibliography as suggested and added the doi on all the references.

Reviewer 2 Report

The manuscript reports findings for a cohort study conducted in Italy on gender differences in celiac disease at diagnosis and approx. 12 months later.

The findings are of some interest, but they could have been much more insightful had statistical methods been used for more than simply getting p-values.

Specific comments:

lines 61-63: The cohort ran over 11 years? This is a long time. How was any shift in patient populations accommodated in the present study? Were effect of year included in the statistical analyses? If not you need to discuss the limitation due to simplistic statistical analyses.

lines 67-69: This inclusion criterion is likely to cause selection bias? Please reflect on whether or not you report findings from a convenience sample, in the discussion under limitations.

lines 147-153: It's not clearly if all specified independent variables were included in the logistic regression model at the same time (a multivariate analysis) or one at a time (a univariate analysis). Please clarify.

line 154: You mean a subgroup analysis!

lines 173-186: Please don't repeat everything significant from the table. Please only report the key findings, so that you indeed provide added value to the content of the table instead of simple repetition. Also, it's somewhat disturbing that you focus on p-values but don't provide any quantification of differences between females and males in terms of estimates and confidence intervals. Not a very informative summary of the data.

Table 2 (and elsewhere): Please report OR's with 2 decimals. Moreover, it's not in all cases clear what the OR means. What units or categories are used for the independent variables. Please provide more details in the table as it should be self-contained.

Author Response

Open Review – REVIEWER 2

(x) I would not like to sign my review report

( ) I would like to sign my review report

English language and style

( ) Extensive editing of English language and style required

( ) Moderate English changes required

( ) English language and style are fine/minor spell check required

(x) I don't feel qualified to judge about the English language and style

Yes         Can be improved             Must be improved          Not applicable

Does the introduction provide sufficient background and include all relevant references?

(x)          ( )           ( )            ( )

Are all the cited references relevant to the research?

(x)          ( )           ( )            ( )

Is the research design appropriate?

(x)          ( )           ( )            ( )

Are the methods adequately described?

( )           ( )           (x)          ( )

Are the results clearly presented?

( )           ( )           (x)          ( )

Are the conclusions supported by the results?

( )           ( )           (x)          ( )

Comments and Suggestions for Authors

The manuscript reports findings for a cohort study conducted in Italy on gender differences in celiac disease at diagnosis and approx. 12 months later.

The findings are of some interest, but they could have been much more insightful had statistical methods been used for more than simply getting p-values.

Specific comments:

lines 61-63: The cohort ran over 11 years? This is a long time. How was any shift in patient populations accommodated in the present study? Were effect of year included in the statistical analyses? If not you need to discuss the limitation due to simplistic statistical analyses.

We thank the Reviewer for the time spent and the attention paid to analyze our work.

We are aware that the inclusion of our  patients covers a long-time period. We decided to consider this period since, starting from 2008, we begin to perform systematically the histologic control after starting the GFD. In addition, the features analyzed during the outpatient visits have been almost the same during these years.

We cannot exclude that some differences could be present. However, we believe that a long period of inclusion is a strength making possible the inclusion of a larger and homogeneous population.

Nevertheless, we are aware that the lack of a statistical analysis evaluating the possible role of time effect could be a limit. For this reason, we included this limitation in the text (Discussion lines 352-355).

lines 67-69: This inclusion criterion is likely to cause selection bias? Please reflect on whether or not you report findings from a convenience sample, in the discussion under limitations.

Thanks for your attention in this topic. In order to limit the inclusion bias in a such wide range of patients we decided to avoid misleading diagnosis and include only patients with both duodenal atrophy and CD-specific antibodies positivity. In fact, in the current Literature, the Marsh I (or the non-atrophic) duodenal histologic lesion has been demonstrated to be secondary to CD in only ¼ of patients (Aziz I, et al. Predictors for Celiac Disease in Adult Cases of Duodenal Intraepithelial Lymphocytosis. J Clin Gastroenterol. 2015, 49,477-82). Concerning the antibodies positivity as inclusion criteria, we are currently aware that a final percentage of 8-10% of patients with atrophic CD were seronegative at the diagnosis time.  

For these reasons, we considered this inclusion criterion to avoid population bias in terms of CD misdiagnosis and, consequently, to include a homogeneous population.   

lines 147-153: It's not clearly if all specified independent variables were included in the logistic regression model at the same time (a multivariate analysis) or one at a time (a univariate analysis). Please clarify.

Thanks for the comment. The independent variables were included in the logistic regression model at the same time as a multivariate analysis (table 2). We included the term ‘Multivariate’ for clarification in the Methods (line 149), in the Results (line 185) and in the title of Table 2. 

 line 154: You mean a subgroup analysis!

We modified the sentence as suggested.

 lines 173-186: Please don't repeat everything significant from the table. Please only report the key findings, so that you indeed provide added value to the content of the table instead of simple repetition. Also, it's somewhat disturbing that you focus on p-values but don't provide any quantification of differences between females and males in terms of estimates and confidence intervals. Not a very informative summary of the data.

We have modified the result section (lines 178-193) in order to make more informative the summary of the data, not merely repeating the content of the cited tables.

Table 2 (and elsewhere): Please report OR's with 2 decimals. Moreover, it's not in all cases clear what the OR means. What units or categories are used for the independent variables. Please provide more details in the table as it should be self-contained.

Thanks for your comment. We added the 2° decimals at the OR’s values both in the Tables and in the text.

We also specified in Table 2 what kind of variables (in terms of units or categories) we used for the Multivariate analysis adding more informative data in the footnotes.
